# Neoadjuvant Concurrent Radiotherapy and Chemotherapy in Early Breast Cancer Patients: Long-Term Results of a Prospective Phase II Trial

**DOI:** 10.3390/cancers13205107

**Published:** 2021-10-12

**Authors:** Diane Jornet, Pierre Loap, Jean-Yves Pierga, Fatima Laki, Anne Vincent-Salomon, Youlia M. Kirova, Alain Fourquet

**Affiliations:** 1Department of Radiation Oncology, Institut Curie, 75005 Paris, France; diane.jornet@curie.fr (D.J.); pierre.loap@curie.fr (P.L.); alain.fourquet@curie.fr (A.F.); 2Department of Medical Oncology, Institut Curie, 75005 Paris, France; jean-yves.pierga@curie.fr; 3Department of Surgical Oncology, Institut Curie, 75005 Paris, France; fatima.laki@curie.fr; 4Department of Pathology, Institut Curie, 75005 Paris, France; anne.salomon@curie.fr; 5Faculty of Medicine, University Versailles St Quentin, 78000 St Quentin, France

**Keywords:** breast cancer, neoadjuvant concurrent radiotherapy and chemotherapy, long term results, prospective phase II

## Abstract

**Simple Summary:**

The originality of this prospective study is to use radiation therapy in association with chemotherapy before surgery and permit patients to preserve their breasts or to undergo immediate reconstruction. This neoadjuvant strategy can therefore allow one-stage breast reconstructive surgery, the so-called “reverse technique”.

**Abstract:**

**Background:** Neoadjuvant concurrent radiochemotherapy makes it possible to increase the breast conservation rate. This study reports the long term outcome of this treatment. **Methods:** From 2001 to 2003, 59 women with T_2–3_ N_0–2_ M_0_ invasive breast cancer (BC) not amenable to upfront breast conserving treatment (BCS) were included in this prospective, non-randomized phase II study. Chemotherapy (CT) consisted of four cycles of continuous 5-FU infusion and Vinorelbine. Starting concurrently with the second CT cycle, normofractionated RT was delivered to the breast and LN. Breast surgery was then performed. **Results:** Median follow-up (FU) was 13 years [3–18]. BCS was performed in 41 (69%) patients and mastectomy in 18 patients, with pathological complete response rate of 27%. Overall and distant-disease free survivals rates at 13 years were 70.9% [95% CI 59.6–84.2] and 71.5% [95% CI 60.5–84.5] respectively. Loco regional and local controls rates were 83.4% [95% CI 73.2–95.0] and 92.1% [95% CI 83.7–100], respectively. Late toxicity (CTCAE-V3) was assessed in 51 patients (86%) with a median follow-up of 13 years. Fifteen presented grade 2 fibrosis (29.4%), 8 (15.7%) had telangiectasia, and 1 had radiodermatitis. **Conclusions:** This combined treatment provided high long-term local control rates with limited side-effects.

## 1. Introduction

The standard treatment for localized breast cancer is lumpectomy followed by radiotherapy to the breast possibly combined with a boost to the tumour bed, or mastectomy when a large tumour prevents breast-conserving treatment. This strategy reduces the risk of local recurrence and improves the patient’s overall survival [1,2]. Avoiding mastectomy is an essential goal for patients in terms of quality of life [3].

In patients who are not eligible for first-line breast-conserving surgery, neoadjuvant chemotherapy has been shown to be effective in terms of downstaging, but to a lesser extent in hormone receptor-positive patients [4]. This neoadjuvant strategy can therefore allow one-stage breast reconstructive surgery, the so-called “reverse technique”. The British non-randomised PRADA trial is currently investigating the feasibility and cosmetic results of this strategy. In addition, no significant difference in terms of overall survival or disease-free survival has been shown between neoadjuvant chemotherapy and adjuvant chemotherapy [5,6]. 

Neoadjuvant radiotherapy, reported in several series, can achieve good complete pathological response rates (6–41%) with good safety at the doses delivered [7]. Other trials reported interesting results of radiochemotherapy [8]. Trial S14 evaluating concomitant neoadjuvant chemoradiotherapy (5FU-Vinorelbine) demonstrated a complete pathological response rate of 27% with acceptable acute toxicity [9,10].

However, there have been considerable advances in breast RT, including intensity modulated RT (IMRT), accelerated partial breast irradiation (APBI), simultaneous integrated boost and (SIB), and image guided radiation (IGRT), that could facilitate preoperative RT. In this modern setting, preoperative RT may be useful in certain situations, including (i) downstaging to enable conservation surgery, (ii) facilitating breast reconstruction, (iii) facilitating partial breast irradiation, and/or (iv) aiding translational research [11].

Herein, we report the 13-year results of trial S14 in terms of long-term toxicities and survival to evaluate these long-term results in the context of modern individualized treatment.

## 2. Materials and Methods

### 2.1. Patients

From November 2001 to September 2003, sixty patients at the Institut Curie with histologically proven, localized, non-inflammatory invasive breast carcinoma, not immediately eligible for breast-conserving surgery were included in the phase II S14 trial.

Patients between the ages of 18 and 65 years had to present a good general condition compatible with concomitant chemoradiotherapy (no history of cancer apart from cervical carcinoma in situ or adequately treated basal cell carcinoma, a Karnofsky performance score greater than 80%, no history of bowel obstruction, severe cardiovascular disease, peripheral neuropathy, and no factors likely to interfere with treatment delivery and monitoring).

A satisfactory laboratory work-up was also required (neutrophils ≥ 1500/mm^3^, platelets ≥ 100,000/mm^3^, haemoglobin ≥ 10 g/dL, total bilirubin ≤ 2 N, aspartate aminotransferase and alanine aminotransferase ≤ 2.5 N, creatinine clearance ≥ 50 mL/min).

Positive diagnosis and staging were based on physical examination, mammogram and bilateral ultrasound, bilateral breast magnetic resonance imaging (MRI), core biopsy, and a standard radiological work-up looking for distant metastases (chest X-ray, abdominal ultrasound, and bone scintigraphy).

The presence of progesterone receptors on the biopsy was initially only investigated in oestrogen receptor-negative patients. HER2 amplification was determined by fluorescence in situ hybridization (FISH) in patients with a HER2 score of 2+ or in the case of heterogeneous immunohistochemical staining.

A new histological diagnosis and IHC hormone receptor analysis was requested in the case of recurrence.

All patients provided their written informed consent, and the Ethics Committee/Institutional Review Board approved the study, which was conducted in accordance with the Declaration of Helsinki and good clinical practice guidelines.

### 2.2. Treatment

In the neoadjuvant setting, patients received four cycles of chemotherapy after port placement with 5FU 500 mg/m^2^/day continuously for five consecutive days and vinorelbine on the first and sixth days, every three weeks, for a total of four cycles. Radiotherapy was started on the first day of the second cycle and delivered according to a normofractionated protocol of 50 Gy in 25 fractions over 5 weeks to the whole breast, and 46 Gy in 23 fractions to the internal mammary chain and supraclavicular and infraclavicular lymph nodes as shown above:**Radiotherapy:****Breast + IM and upper axillary nodes   50 Gy/25 f****Chemotherapy**5FU iv 24 h infusion        500 mg/m^2^, D1 to D5Vinorelbine IV         25 mg/m^2^, D1 and D54 cycles, 21 days each**Radiotherapy started concurrently with CT #2**

At least six weeks after completion of radiotherapy, patients then underwent breast-conserving or non-conservative surgery, depending on the response to neoadjuvant therapy, and Berg level I and II axillary lymph node dissection.

Adjuvant chemotherapy consisting of four cycles of 5-FU, Epirubicin, Cyclophosphamide (FEC) was proposed in the presence of factors of poor prognosis, age, and absence of complete pathological response. A boost of 16 Gy/8 fractions to the tumour bed was proposed according to Institut Curie guidelines in patients with BCS without complete pathological response. Hormone receptor-positive patients received tamoxifen therapy for a minimum of five years (followed by a switch to exemestane in postmenopausal patients after marketing authorisation had been granted).

### 2.3. Evaluation

All data were collected retrospectively from consultation reports using the Institut Curie’s medical data software.

Population characteristics were extracted from individual data and not reported from previous analyses.

Cosmetic results (pigmentation, telangiectasia, fibrosis) were analysed when an evaluation was available five or ten years after completion of radiotherapy. These results were extracted from oncologist-radiotherapist consultation reports and graded according to CTCAE Common Terminology Criteria for Adverse Events Version 5.0 (CTCAE). The presence of cardiovascular and thyroid events was also investigated.

Clinical follow-up was ensured every six months, with annual bilateral mammogram looking for possible recurrence, defined as local, locoregional, in-field, distant recurrence, or second cancer.

### 2.4. Statistical Analysis

Survival was defined as the time from surgery until occurrence of the event. For locoregional and distant recurrence-free survival and overall survival, patients were censored at the date of last known contact.

Survival and interval rates were calculated by the Kaplan-Meier method, and groups were compared using a log-rank test. Non-adjusted hazard ratios were performed using the Cox proportional model and multivariate analysis was carried out to assess the adjusted influence of prognostic factors using the Cox stepwise procedure.

The covariates selected for multivariate analysis were those with a *p*-value less or equal to 0.25 on univariate analysis.

The limit of significance was *p* < 0.05. Statistical analyses were performed using R software version 3.6.1.

## 3. Results

### 3.1. Patient Characteristics

One of the sixty patients included withdrew her consent.

All patients and tumours’ characteristics described are summarized in Table 1.

The study population comprised women with a mean age of 49 years [31–65], 41% of whom were postmenopausal. Pre-treatment MRI reported lesions with a mean long axis of 38 mm (20–80 mm), corresponding to cT2 lesions in 73% of cases and cT3 lesions in 27% of cases.

Histology mainly consisted of ductal (68%) and lobular (22%) carcinomas, followed by a few rarer histological subtypes (10%), 31% of grade SBR 3, and 49% of grade SBR 2. Eight tumours also presented HER2 overexpression with positive hormone receptors in 75% of cases.

### 3.2. Treatment Characteristics

All treatments are reported in Table 2.

### 3.3. Safety of Treatment

No grade 4 or 5 toxicity was reported and only one grade 3 adverse reaction was reported more than 5 years after treatment.

Among the 51 (86%) patients with median 13 years of follow-up, grade 2 fibrosis was reported in 15 patients (29%), grade 2 telangiectasia was reported in 7 patients (14%), and grade 3 telangiectasia was reported in 1 patient; and grade 2 radiodermatitis was reported in only one patient.

Three patients developed a cardiac arrhythmia requiring effective anticoagulation. These patients were treated for right breast cancer in two cases and left breast cancer in one case. Finally, one case of hypothyroidism was diagnosed.

### 3.4. Response to Treatment

As previously described, 16 patients obtained a complete pathological response (defined as residual malignant epithelial cells representing <5% of the initial tumour mass with no detectable mitotic figures, or exclusively in situ carcinoma) including three with localized ductal carcinoma in situ. Total mastectomy was able to be avoided in 69% of these patients (Table 2). Berg level I and II lymph node dissection removed a median of 11 lymph nodes (range: 3–23).

Adjuvant therapy consisted of FEC 100 chemotherapy alone in 17% of cases, hormonal therapy alone in 20% of cases, and a combination of chemotherapy and hormonal therapy in 51% of cases (Table 2).

### 3.5. Survival Data

Overall survival at 13 years was 70.9% (95%CI: 59.6–84.2) (Figure 1). Twenty patients had died, including 17 patients who died from breast cancer. Univariate analysis revealed a correlation between histological grade and overall survival (*p* = 0.01), which was confirmed by multivariate analysis (*p* = 0.04) (Table 3).

Metastasis-free survival was 71.5% (95%CI: 60.5–84.5) at 13 years (Figure 2) and 19 patients presented metastatic progression. Four patients developed a second cancer (acute myeloid leukaemia, ovarian cancer, squamous cell carcinoma, and lung cancer contralateral to the primary breast tumour). Three patients also developed contralateral infiltrating carcinoma.

The locoregional control rate was 83.4% (95%CI: 73.2–95.0) (Figure 3): two breast and regional recurrences, and five regional recurrences alone. All regional recurrences were associated with metastatic progression, except for one isolated regional recurrence (axillary and supraclavicular recurrence managed by locoregional surgery and radiotherapy and systemic therapy, with stable disease 13 years after the recurrence).

With a median follow-up of 13 years (3–18 years), the local control rate was 92.1% (95%CI: 83.7–100) (Figure 4): as following, two ductal carcinomas were reported 7 and 10 years after discontinuation of hormonal therapy and one ductal carcinoma in situ was reported more than 10 years after the end of treatment. Each of these recurrences was biopsied. The initial lesions were grade 1, HR+, HER2- invasive ductal carcinomas, one of which had been treated by mastectomy and a triple-negative undifferentiated carcinoma.

## 4. Discussion

Review of the data from this phase II study allows assessment of the safety and efficacy of 5FU-Vinorelbine regiment used concurrently with radiotherapy with a follow-up of 13 years in this prospective study of homogeneous cohort of patients with localized breast cancer not eligible for first-line breast-conserving surgery. The local control rate of 92.1% was very satisfactory compared with historical series (7, 8, 11). Regional recurrences mainly related to regional recurrence alone (5/7, 71%), which is quite unusual and could be explained by the absence of lower axillary radiotherapy.

Histological grade is a prognostic factor for overall survival, but the small sample sizes do not allow sufficient power to assess the role of other prognostic factors.

In women treated by breast-conserving surgery in this study, the chronic toxicity was acceptable and encouraging, in line with the previously reported acute and late toxicity [9,10,11]. Note that radiotherapy consisted of 3D conformal radiotherapy using a cobalt source. The toxicities reported in this study do not constitute a limiting factor for the growth of neoadjuvant radiotherapy, as modern radiotherapy techniques allow much better results by acting on several technical factors, such as the use of photon beams or even protons [12], 3D conformal radiotherapy or IMRT, an integrated boost dose [13], the lateral decubitus position [14], and respiratory gating [15].

This neoadjuvant strategy could also have the advantage of improving cosmetic results in the context of breast reconstruction [16]. 

A number of older case series and single arm trials report on preoperative RT with or without concomitant chemotherapy [17,18,19,20,21,22,23,24,25,26,27]. In those that report on receptor status, hormone receptor positive tumours were less likely to achieve pathological complete response to chemoradiation (chemoRT) than other subtypes [25,26], which is unsurprising given the better complete pathological response rates following chemotherapy for higher risk subgroups. Those reporting on complications in general found more acute toxicity than would be expected with modern postoperative breast RT.

Several hypofractionated radiotherapy regimens have been shown to be non-inferior to conventional regimens in terms of recurrence and decreased toxicity [28,29,30]. Many clinical trials are currently studying the feasibility of hypofractionated radiotherapy or neoadjuvant SBRT [31] to allow downstaging and are also testing the hypothesis of an immune-induced response that could prolong overall survival or DFS.

Target volumes are also under review with the initiation of clinical trials of accelerated partial breast irradiation, especially in small lesions [32]. In particular, we are waiting for the results of the Neo-APBI-01 randomized trial (RCB ID No.: 2015-A01062-47), which is comparing neoadjuvant chemotherapy and accelerated partial breast irradiation to neoadjuvant chemotherapy alone in patients with triple-negative and Luminal B breast cancer ineligible for first-line breast-conserving surgery. The results of trials of neoadjuvant radiotherapy combined with targeted therapies [33] or immunotherapy are also eagerly awaited.

## 5. Conclusions

Neoadjuvant concurrent radio chemotherapy makes it possible to increase the breast conservation rate. This combined treatment modality provided high long-term local control rates with limited side-effects. Further prospective larger studies focusing on modern radiotherapy techniques and new combinations of chemotherapy or targeted therapy are needed to improve the results.

## Figures and Tables

**Figure 1 cancers-13-05107-f001:**
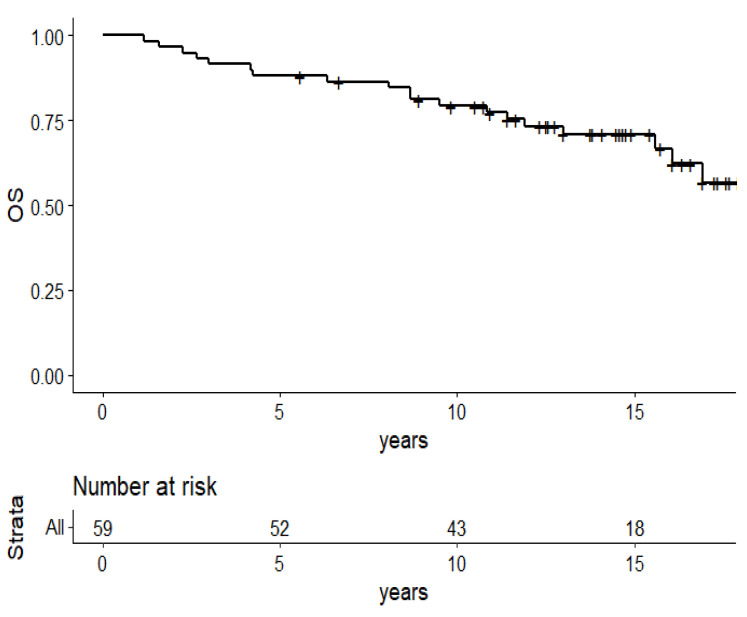
Kaplan–Meier survival plots of the 59 patients: overall survival.

**Figure 2 cancers-13-05107-f002:**
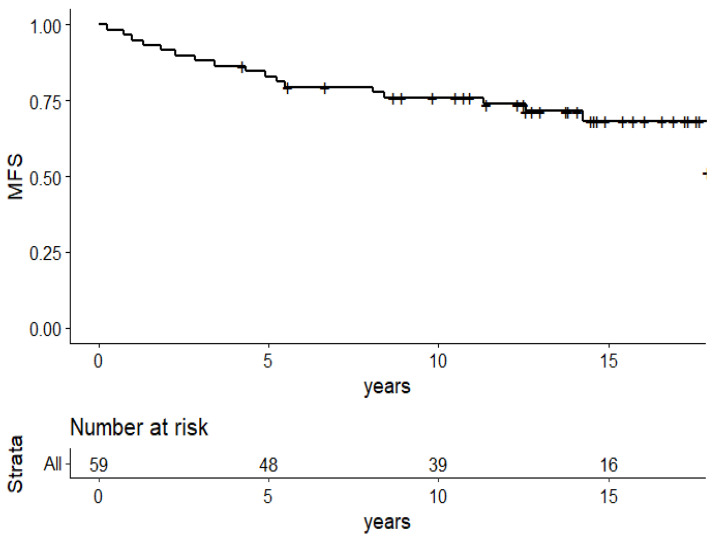
Kaplan–Meier survival plots of the 59 patients: metastasis free survival.

**Figure 3 cancers-13-05107-f003:**
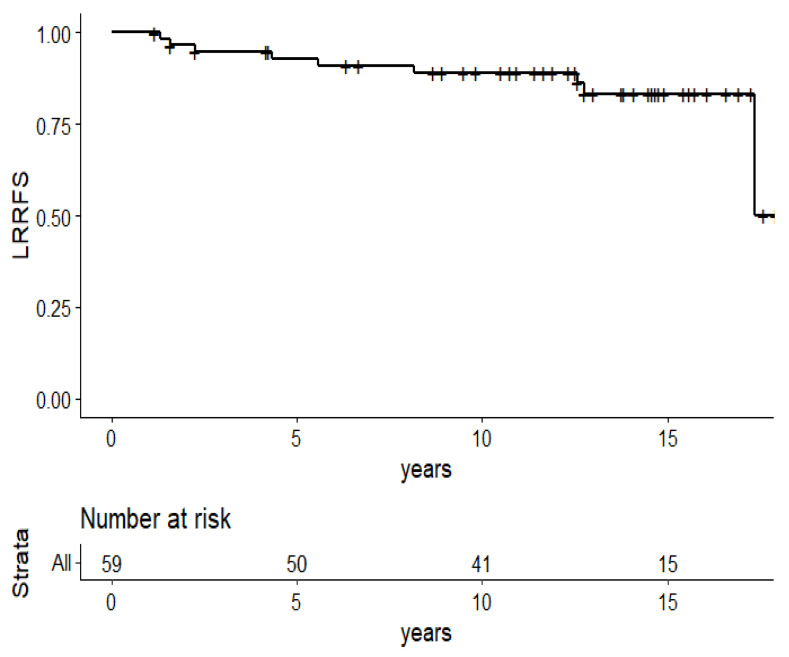
Kaplan–Meier survival plots of the 59 patients: locoregional relapse free survival.

**Figure 4 cancers-13-05107-f004:**
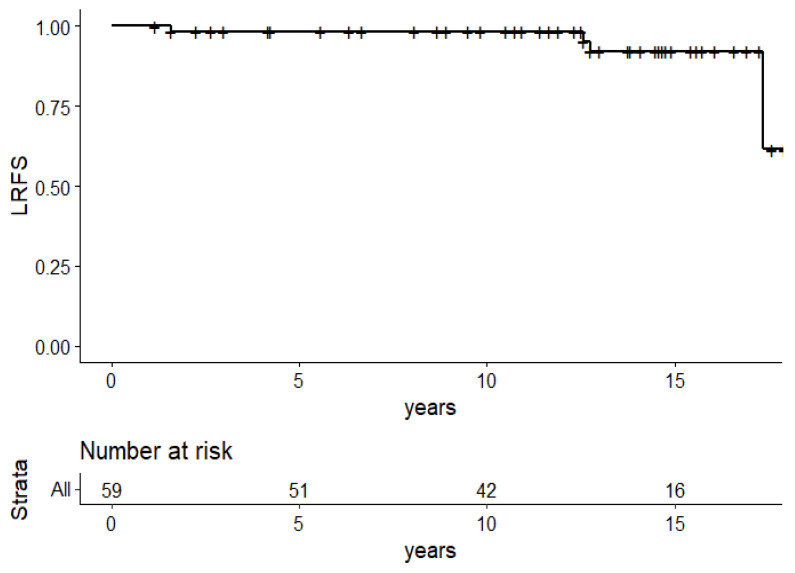
Kaplan–Meier survival plots of the 59 patients: local relapse free survival.

**Table 1 cancers-13-05107-t001:** Patient and tumour characteristics.

Characteristics		*n*	%
Age—years (median [range])	49 (31–65)		
*Menopausal*			
Yes		24	41
No		35	59
Tumor maximal diameter (Baseline MRI)—mm (median [range])	38 (20–80)		
*Clinical stage*			
T2N0		26	44
T2N1		17	29
T3N0		9	15
T3N1		7	12
*Infiltrating carcinoma*			
Ductal		40	68
Lobular		13	22
Other		6	10
*Histological grade*			
1		12	20
2		29	49
3		18	31
*Number of mitoses/10 high power field*			
0		4	7
<11		38	64
11–22		4	7
>22		13	22
*HER2 over-expression*			
Yes		8	14
No		51	86
*Ductal carcinoma* in situ *component*			
Yes		21	36
No		38	64
*Estrogen/progesterone receptors*			
ER+/PR unknown		4	7
ER+/PR+		9	15
ER+/PR-		27	46
ER-/PR+		4	7
ER-/PR-		15	25

**Table 2 cancers-13-05107-t002:** Treatment characteristics.

		*n*	%
*Surgery*			
Breast conserving surgery		41	69
Mastectomy		18	31
*Final surgical margin (infiltrating or DCIS)*			
Minimal involvement		8	14
Extensive involvement		2	3
No involvement		49	83
Number of axillary lymph nodes dissected: median [min-max]	11 (3–23)		
*Radiotherapy*			
Dose to the breast—Gy: median [min–max]	50 (46–52)		
Boost dose to the tumorectomy bed—Gy: median [min–max]	16 (5, 4–26)		
*Radiotherapy boost after breast conserving surgery (n = 40)*			
Yes		38	93
No		3	7
*Adjuvant systemic treatment*			
None		7	12
Chemotherapy alone		10	17
Hormone-therapy alone		12	20
Chemo- and hormone-therapy		30	51

**Table 3 cancers-13-05107-t003:** Univariate and multivariate analyses of prognostic factors for locoregional recurrences, metastasis free survival, and overall survival.

	*n*	13-Year LRRFS (95% CI)	*p*-Value of Log Rank Test	Relative Risk [95% CI]	13-Year MFS (95% CI)	*p*-Value of log Rank Test	Relative Risk [95% CI]	13-Year OS (95% CI)	*p*-Value of log Rank Test	Relative Risk [95% CI]
Body mass index	MD = 2		0.97	1 [0.87–1.16]		0.68	1.02 [0.92–1.14]		0.64	1.03 [0.92–1.14]
Age	MD = 0		0.77	0.73 [0.09–5.87]		0.49	0.49 [0.06–3.7]		0.43	0.44 [0.06–3.34]
<40 years	6	83 (58–100)			83 (58–100)			83 (58–100)		
≥40 years	53	83 (72–96)			70 (58–84)			70 (58–84)		
Clinical T stage	MD = 0		0.66	1.36 [0.35–5.27]		0.19	1.89 [0.73–4.89]		0.82	1.12 [0.43–2.92]
T2	43	86 (75–98)			76 (64–90)			73 (60–88)		
T3	16	75 (53–100)			58 (37–92)			66 (46–96)		
Clinical N stage	MD = 0		0.15	2.56 [0.72–9.14]		0.19	1.82 [0.74–4.51]		0.18	1.83 [0.76–4.43]
N0	35	91 (79–100)			76 (62–92)			79 (67–94)		
N1–N2	24	72 (56–94)			66 (49–88)			58 (40–84)		
Histological type	MD = 0		0.63	1.4 [0.36–5.43]		0.36	0.65 [0.26–1.62]		0.67	0.82 [0.34–2.02]
Ductal	40	81 (68–97)			74 (62–89)			71 (57–87)		
Non-ductal	19	87 (72–100)			68 (50–93)			74 (56–96)		
Histological grade	MD = 0		0.38	1.9 [0.45–7.99]		0.11	2.21 [0.84–5.82]		0.01	3.3 [1.31–8.3]
1 or 2	41	85 (73–98)			77 (64–91)			78 (66–93)		
3	18	82 (65–100)			61 (42–88)			54 (35–84)		
Hormone receptors	MD = 0		0.17	2.46 [0.69–8.8]		0.49	1.44 [0.51–4.1]		0.33	1.61 [0.61–4.26]
Negative	15	79 (60–100)			67 (47–95)			58 (37–91)		
Positive	44	84 (72–99)			73 (60–88)			76 (63–90)		
Her2 over-expression	MD = 0		0.14	2.84 [0.71–11.31]		0.49	1.55 [0.44–5.42]		0.55	1.46 [0.42–5.04]
Yes	8	62 (37–100)			62 (37–100)			54 (26–100)		
No	51	87 (77–99)			73 (62–87)			73 (61–87)		
Triple negative	MD = 0		0.37	1.86 [0.48–7.27]		0.58	1.38 [0.45–4.24]		0.36	1.62 [0.58–4.51]
Yes	12	82 (62–100)			67 (45–99)			58 (36–94)		
No	47	83 (71–97)			72 (60–87)			75 (63–89)		
Number of mitoses/10 HPF	MD = 0		0.38	0.39 [0.05–3.17]		0.38	1.56 [0.57–4.21]		0.18	1.92 [0.74–4.97]
<11	42	80 (67–95)			74 (61–90)			74 (61–89)		
≥11	17	94 (83–100)			65 (46–92)			64 (45–92)		
<44	35	84 (70–100)			73 (60–90)			70 (55–88)		
≥44	23	81 (66–100)			69 (52–91)			72 (55–94)		
Yes	53	84 (73–96)			72 (61–86)			72 (60–86)		
No	6	75 (43–100)			60 (29–100)			60 (29–100)		
pCR	MD = 0		0.78	0.83 [0.21–3.21]		0.63	1.31 [0.43–3.98]		0.82	1.12 [0.41–3.1]
Yes	16	80 (62–100)			81 (64–100)			80 (62–100)		
No	43	83 (71–99)			68 (55–84)			67 (54–84)		

MD = missing data.

## Data Availability

In case of additional questions, the data are available.

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
