# Peer review of "Neoadjuvant Concurrent Radiotherapy and Chemotherapy in Early Breast Cancer Patients: Long-Term Results of a Prospective Phase II Trial"

_cancers, 2021, doi:10.3390/cancers13205107_

Round 1

Reviewer 1 Report

I found the paper to be overall well written and felt confident that the authors performed careful and thorough analysis. The strength of the study is its long follow-up time.
Some comments

1) In the introduction, it would be helpful to explain how this study adds to current knowledge and challenge existing paradigms? Please indicate any relevant pieces of previous work done under similar cohort

2) Discuss a bit about success or failure of other chemo ( xeloda/taxanes...etc) 

3) Any potential limitations of the study ? (eg. small sample size? ...)

Author Response

RESPONSES to Reviewers Point by Point :

Reviewer 1

I found the paper to be overall well written and felt confident that the authors performed careful and thorough analysis. The strength of the study is its long follow-up time.

The authors are thankful to reviewer for this positive remark of this work.

Some comments

  • In the introduction, it would be helpful to explain how this study adds to current knowledge and challenge existing paradigms? Please indicate any relevant pieces of previous work done under similar cohort

The authors are thankful to reviewer for this suggestion. Changes were done, in the text:

Neoadjuvant radiotherapy, reported in several series, can achieve good complete pathological response rates (6-41%) with good safety at the doses delivered. [7]. Other trials reported interesting results of radiochemotherapy [8]. Trial S14 evaluating concomitant neoadjuvant chemoradiotherapy (5FU-Vinorelbine) demonstrated a complete pathological response rate of 27% with acceptable acute toxicity. [9, 10]

However, there have been considerable advances in breast RT, including intensity modulated RT (IMRT), accelerated partial breast irradiation (APBI), simultaneous integrated boost and (SIB) and image guided radiation (IGRT) that could facilitate preoperative RT. In this modern setting, preoperative RT may be useful in certain situations, which are discussed: (i) downstaging to enable conservation surgery, (ii) facilitating breast reconstruction, (iii) facilitating partial breast irradiation, and (iv) aiding translational research [11].

Herein, we report the 13-year results of trial S14 in terms of long-term toxicities and survival to evaluate these long term results in the context of modern individualized treatment.

  • Discuss a bit about success or failure of other chemo ( xeloda/taxanes...etc) 

We added numerous references and in the discussion:

A number of older case series and single arm trials report on preoperative RT with or without concomitant chemotherapy[17–27]. In those that report on receptor status, hormone receptor positive tumours were less likely to achieve pathological complete response to chemoradiation (chemoRT) than other subtypes[25, 26], which is unsurprising given the better complete pathological response rates following chemotherapy for higher risk subgroups. Those reporting on complications in general found more acute toxicity than would be expected with modern postoperative breast RT.

  • Any potential limitations of the study ? (eg. small sample size? ...)

The study was designed for this number of patients but we added:

Histological grade is a prognostic factor for overall survival, but the small sample sizes do not allow sufficient power to assess the role of other prognostic factors.

Reviewer 2 Report

The study addressed the effectiveness of concurrent radio- chemotherapy approach for breast cancer treatment and breast preservation. Authors investigated the long term outcomes of this approach. The study is very interesting and comprehensive. However, there are several problems to address. The most important point: some conclusions were not supported by data.

  1. English language editing is required to improve the clarity of this work.

For instance, the 1st sentence (lines 14-15) should be corrected (“… permit patients to preserve their breast or undergo the immediate reconstruction…”).

  1. It is not acceptable to use ‘pts’ instead of “patients’’ in the abstract; do not use ‘resp”, use “respectively”.
  2. Introduction: the so-called “reverse techniques’ was not introduced properly and requires more detailed description. Why it is considered beneficial? Any previous studies demonstrated this?
  3. The treatment method (lines 81-onwards) can be described using a diagram. It will visualize the process and may attract attention of larger groups of readers.
  4. Table 2 should be visually improved; all numbers should be aligned.
  5. Table 3 should be re-organized and all boarders ( for each column) should be provided as it is very difficult to read the current version.
  6. Line 197: it is not “large” cohort. Please remove. You contradicted yourselves ( line 202: “…small sample size…”)
  7. Line 198: you cannot make this conclusion as you have not presented data comparison.  With what "historical series" did you compare your data? You have to show the comparisons (graphs etc?).
  8. Line 204: how the toxicity was assessed? Any data to confirm your words? You have to describe the methods and collected data to support this statement. 

Author Response

The study addressed the effectiveness of concurrent radio- chemotherapy approach for breast cancer treatment and breast preservation. Authors investigated the long term outcomes of this approach. The study is very interesting and comprehensive. However, there are several problems to address. The most important point: some conclusions were not supported by data.

The authors are thankful to reviewer for this positive remark of this work.

  1. English language editing is required to improve the clarity of this work.

For instance, the 1st sentence (lines 14-15) should be corrected (“… permit patients to preserve their breast or undergo the immediate reconstruction…”).

Done, the English was corrected by professional medical translator, in the text:

The originality of this prospective study is to use the radiation therapy in association with chemotherapy before the surgery and permit to patients to preserve their breasts or to undergo the immediate reconstruction.

  1. It is not acceptable to use ‘pts’ instead of “patients’’ in the abstract; do not use ‘resp”, use “respectively”.

Corrected.

  1. Introduction: the so-called “reverse techniques’ was not introduced properly and requires more detailed description. Why it is considered beneficial? Any previous studies demonstrated this?

The authors are agreeing with the reviewer, therefore they added the information:

The British non-randomised PRADA trial is currently investigating the feasibility and cosmetic results of this strategy.

  1. The treatment method (lines 81-onwards) can be described using a diagram. It will visualize the process and may attract attention of larger groups of readers. Done, visual support was added.

Radiotherapy:

           Breast + IM and upper axillary nodes     50 Gy / 25 f

Chemotherapy

5FU iv 24h infusion                                   500 mg / m2, D1 to D5

Vinorelbine IV                                              25 mg / m2, D1 and D5

  • cycles, 21 days each

Radiotherapy started concurrently with CT #2

  1. Table 2 should be visually improved; all numbers should be aligned.

Done

  1. Table 3 should be re-organized and all boarders ( for each column) should be provided as it is very difficult to read the current version.

The improvements were realized, and then it depends of Editorial team how it will be in the published version.

  1. Line 197: it is not “large” cohort. Please remove. You contradicted yourselves ( line 202: “…small sample size…”)

Done, changed in the text:

…in this prospective study of homogeneous cohort of patients

  1. Line 198: you cannot make this conclusion as you have not presented data comparison.  With what "historical series" did you compare your data? You have to show the comparisons (graphs etc?).

Done, references were added:

The local control rate of 92.1% was very satisfactory compared with historical series (7, 8, 11).

  1. Line 204: how the toxicity was assessed? Any data to confirm your words? You have to describe the methods and collected data to support this statement. 

Done, changed in the text (Methods and Materials):

Cosmetic results (pigmentation, telangiectasia, fibrosis) were analysed when an evaluation was available five or ten years after completion of radiotherapy. These results were extracted from oncologist-radiotherapist consultation reports and graded according to CTCAE Common Terminology Criteria for Adverse Events Version 5.0 (CTCAE). The presence of cardiovascular and thyroid events was also investigated.

Clinical follow-up was ensured every six months, with annual bilateral mammogram looking for possible recurrence, defined as local, locoregional, in-field, distant recurrence or second cancer.

Done, references were added in Discussion:

In women treated by breast-conserving surgery in this study, the chronic toxicity was acceptable and encouraging, in line with the previously reported acute and late toxicity (9, 10).

Reviewer 3 Report

The objective of the study was to review the data from Phase II study and provide long term outcomes of concurrent radiotherapy and chemotherapy (5FU-Vinorelbine regimen) in early breast cancer patients (who are not eligible for first line breast conserving surgery {BCS}). The reported findings are from median follow-up period of 13 years (3-18 years). BCS was performed in 41 patients and mastectomy was performed in 18 patients. The authors have reported overall and distant disease free survival rates at 13 years as 70.9% and 71.5% respectively. The authors have also reported locoregional and local control rates as 83.4% and 92.1% respectively. The authors have reported high long term local control rates with limited side effects. The authors have previously published data on the outcomes of this study with findings on 5 year follow-up (5-7 years) in 2012 in Radiotherapy and Oncology and the data presented here is with findings on 13 year follow-up. The data is interesting but the efficacy of preoperative concurrent administration of radiotherapy and chemotherapy needs to be evaluated in large cohort study. 

Minor concerns:

1. Could authors please cross check and correctly cite the references. The authors must have meant to cite the reference # 9 but have cited reference # 8 on Ln # 51 in the introduction.

2. Could authors please cross check and correct the sample size mentioned under "Radiotherapy boost after breast conserving surgery" in Table 2? The sample size should have been 41 as per number of patients who had breast conserving surgery but it is mentioned as 40. Please look into it and correct it.

Author Response

The objective of the study was to review the data from Phase II study and provide long term outcomes of concurrent radiotherapy and chemotherapy (5FU-Vinorelbine regimen) in early breast cancer patients (who are not eligible for first line breast conserving surgery {BCS}). The reported findings are from median follow-up period of 13 years (3-18 years). BCS was performed in 41 patients and mastectomy was performed in 18 patients. The authors have reported overall and distant disease free survival rates at 13 years as 70.9% and 71.5% respectively. The authors have also reported locoregional and local control rates as 83.4% and 92.1% respectively. The authors have reported high long term local control rates with limited side effects. The authors have previously published data on the outcomes of this study with findings on 5 year follow-up (5-7 years) in 2012 in Radiotherapy and Oncology and the data presented here is with findings on 13 year follow-up. The data is interesting but the efficacy of preoperative concurrent administration of radiotherapy and chemotherapy needs to be evaluated in large cohort study. 

The authors are thankful to reviewer for this good understanding and knowledge of our work and we are agreeing that the efficacy of preoperative concurrent administration of radiotherapy and chemotherapy needs to be evaluated in large cohort study. Added in the conclusions:

Further prospective larger studies focusing on modern radiotherapy techniques and new combinations of chemotherapy or targeted therapy are needed to improve the results.

Minor concerns:

  1. Could authors please cross check and correctly cite the references. The authors must have meant to cite the reference # 9 but have cited reference # 8 on Ln # 51 in the introduction.

Done, corrected:

Neoadjuvant radiotherapy, reported in several series, can achieve good complete pathological response rates (6-41%) with good safety at the doses delivered. [7]. Other trials reported interesting results of radiochemotherapy [8]. Trial S14 evaluating concomitant neoadjuvant chemoradiotherapy (5FU-Vinorelbine) demonstrated a complete pathological response rate of 27% with acceptable acute toxicity. [9, 10]

  1. Could authors please cross check and correct the sample size mentioned under "Radiotherapy boost after breast conserving surgery" in Table 2? The sample size should have been 41 as per number of patients who had breast conserving surgery but it is mentioned as 40. Please look into it and correct it.

Done, it is :

Breast conserving surgery

41

Round 2

Reviewer 2 Report

I am satisfied with the revised version of the manuscript. Authors addressed all my comments properly.